# Synergistic Effect of Acetazolamide-(2-hydroxy)propyl β-Cyclodextrin in Timolol Liposomes for Decreasing and Prolonging Intraocular Pressure Levels

**DOI:** 10.3390/pharmaceutics13122010

**Published:** 2021-11-25

**Authors:** Carmen M. Arroyo-García, Daniela Quinteros, Santiago D. Palma, Cesáreo J. Jiménez de los Santos, José R. Moyano, Antonio M. Rabasco, María Luisa González-Rodríguez

**Affiliations:** 1Department of Pharmacy and Pharmaceutical Technology, Faculty of Pharmacy, Universidad de Sevilla, C/Prof. García González, 2, 41012 Sevilla, Spain; carmarroyogarcia@gmail.com (C.M.A.-G.); cesar_1bmv@hotmail.com (C.J.J.d.l.S.); jrmoyano@us.es (J.R.M.); amra@us.es (A.M.R.); 2Unidad de Investigación y Desarrollo en Tecnología Farmacéutica (UNITEFA), CONICET, Universidad Nacional de Córdoba, Ciudad Universitaria, Córdoba 5000, Argentina; danielaquinteros@unc.edu.ar (D.Q.); sdpalma@unc.edu.ar (S.D.P.); 3Departamento de Farmacia, Facultad de Ciencias Químicas, Universidad Nacional de Córdoba, Ciudad Universitaria, Córdoba 5000, Argentina

**Keywords:** liposome, glaucoma, co-loading, timolol, acetazolamide, design of experiments, intraocular pressure, (2-hydroxy)propyl β-cyclodextrin, cyclodextrin competence, ^31^PNMR, drug delivery

## Abstract

The purpose of this study was to design, for the first time, a co-loaded liposomal formulation (CLL) for treatment of glaucoma including timolol maleate (TM) in the lipid bilayer and acetazolamide (Acz)-(2-hydroxy)propyl β-cyclodextrin (HPβCD) complexes (AczHP) solubilized in the aqueous core of liposomes. Formulations with TM (TM-L) and AczHP (AczHP-L), separately, were also prepared and characterized. A preliminary study comprising the Acz/HPβCD complexes and their interaction with cholesterol (a component of the lipid bilayer) was realized. Then, a screening study on formulation factors affecting the quality of the product was carried out following the design of the experiment methodology. In addition, in vitro release and permeation studies and in vivo lowering intraocular pressure (IOP) studies were performed. The results of the inclusion complexation behavior, characterization, and binding ability of Acz with HPβCD showed that HPβCD could enhance the water solubility of Acz despite the weak binding ability of the complex. Ch disturbed the stability and solubility parameters of Acz due to the fact of its competence by CD; thus, Chems (steroid derivative) was selected for further liposome formulation studies. The optimization of the lipid bilayer composition (DDAB, 0.0173 mmol and no double loading) and the extrusion as methods to reduce vesicle size were crucial for improving the physico-chemical properties and encapsulation efficiency of both drugs. In vitro release and permeation studies demonstrated that the CLL formulation showed improvement in in vitro drug release and permeation compared to the liposomal formulations with a single drug (TM-L and AczHP-L) and the standard solutions (TM-S and AczHP-S). CLL showed high efficacy in reducing and prolonging IOP, suggesting that the synergistic effect of TM and Acz on aqueous humor retention and the presence of this cyclodextrin and liposomes as permeation enhancers are responsible for the success of this strategy of co-loading for glaucoma therapy.

## 1. Introduction

Glaucoma is a chronic, progressive, and irreversible optic neuropathy characterized by the progressive loss of retinal ganglion cells. As a result, there is a decrease in the number of axons derived from these cells with the consequent atrophy of the optic nerve [1]. Increased intraocular pressure (IOP) is the major clinically modifiable risk factor for glaucoma progression (vision loss) caused by the imbalance of aqueous humor production and outflow. Topical administration continues to be the most common therapeutic strategy for reducing IOP.

Currently, the pharmacological treatment includes molecules, such as beta-blockers, alpha-2 adrenergic agonists, carbonic anhydrase inhibitors, muscarinic agonists, prostaglandins, nitric oxide donating agents [2], neuroprotective agents [3], and rho-kinase inhibitors [4]. These therapeutic agents, either alone or in combination, are effective in reducing IOP [5]. Recently, RNA-interference-based mechanisms, especially small interfering RNAs (siRNAs), have been under research by scientists for the treatment of glaucoma [6].

Usually, eye drops require multiple daily doses (except for prostaglandins which require once-a-day dosing) due to the partial loss of drug formulation after instillation. This leads to pulsatile time-courses of drug concentrations, which could result in elevated IOP at different points during a day. This inconvenience could be avoided with the use of effective sustained release drug delivery systems for long-term treatment to improve patient adherence and drug availability. Among the numerous strategies [7], the use of nanoparticulate systems has greatly contributed to improving the therapeutic efficacy of ocular diseases [8,9].

Among them, liposomes have emerged as desirable drug carriers for ophthalmic delivery because of their good biocompatibility, biodegradability, and cell membrane-like structure, enabling the drugs to bypass several biological barriers such as the cornea [10].

Working to improve the antiglaucomatous therapy and taking advantage of liposomes to encapsulate both hydrophobic and hydrophilic molecules, many efforts have been made to co-encapsulate two different drugs having a synergistic effect on glaucoma treatment, searching for an increased efficacy. In the ophthalmic field, fewer studies have been realized with liposomes. Dubey et al. [11] developed a latanoprost–timolol maleate fixed combination in liposomes, having the advantage of a more convenient delivery system as well as providing a simple summing effect of the two constituents. Fahmy et al. [12] demonstrated the effectiveness of liposomal carriers of latanoprost and thymoquinone in lowering the IOP upon subconjunctival injection, extending the study to evaluate the effect on the aqueous humor oxidative stress.

Timolol maleate (TM) is a non-selective beta-blocker drug used in the treatment of glaucoma for more than three decades [13]. The antiglaucomatous efficacy of this molecule has been widely demonstrated, as it decreases the IOP by reducing the aqueous humor fluid by blocking the beta receptors in the ciliary body [14]. However, TM from marketed ocular products has several drawbacks such as low bioavailability, frequent instillation, and concomitant patient non-compliance. In addition, some patients manifest a few side effects including local irritation, dryness, conjunctival hyperemia, blurring, and stinging [9]. Other systemic effects comprise bradycardia, hypotension, and the cardiovascular effects of β-antagonist adrenergic drugs [15]. Several studies showed the existence of a significant percentage of atrioventricular blocks caused by beta-blocker eye drops [16,17].

TM formulations have been studied for prolonging the contact time on the ocular surface and increasing the corneal permeability of drugs. For this, several strategies for drug delivery have been exploited, alone or combined: in situ gelling systems [18,19,20]; contact lenses [21]; suspensions, ointments, and nanoparticles in contact lenses [22]; ocular films [23]; liposomes in gelling systems [1,24], nanoparticles [25], liposomes [26], gelatinized core liposomes [27], and emulsions.

In a previous study, TM-loaded liposomes were compared with TM eye drops for their efficacy in reducing IOP in normotensive rabbits. This study indicated a similar IOP reduction (23% IOP reduction, approximately) in rabbits at the end of 7 days after applying a liposomal delivery system (0.5 mg/mL) and 5 mg/mL TM eye drops [28]. Therefore, the use of a lower TM concentration than marketed eye drops strongly enhanced the pharmacological and toxicological profile of TM from liposomal ophthalmic formulations.

In an attempt to better optimize the ophthalmic therapies with TM and exploit the success of co-administering two or more drugs (having synergistic effect on glaucoma) in a single formulation [29,30,31,32], a co-loading drug delivery strategy has also been explored. Although some evidence exists in the literature, for example nanosheet [17] and micelle co-loading TM with latanoprost [33], however, fewer works with liposomes have been found.

Acetazolamide (Acz) is a carbonic anhydrase inhibitor with a potent specific effect in reducing IOP, since it decreases the aqueous humor production. To date, it is still systemically the most effective and highly prescribed drug for the treatment of open-angle glaucoma. Unfortunately, its topical instillation as eye drops is not effective, because this drug belongs to class IV according to the biopharmaceutical classification system (BCS), exhibiting low water solubility (<1 mg/mL) and a low permeability coefficient 4.1 × 10^−6^ cm/s [34]. In addition, it has been reported that Acz is highly unstable at alkaline pH, and at this pH, the molecule occurs in the ionized form, which may further limit its transport through the corneal lipophilic epithelium [35].

Many technological approaches have been investigated to increase Acz solubility and enhance its permeability through the corneal surface such as dendrimers [36], nanoliposomes [37], nasones [38], cyclodextrins [39], polymeric nanoparticles [40], bilosomes [41], microsponge in situ gels [42], nanoemulsions [43], and cocrystals [44].

CDs are natural cyclic oligosaccharides widely used as solubility enhancers of poorly soluble drugs. The toroidal conformation (truncated cone) of their monomers arranges them in a way that delimits a relatively apolar hydrophobic cavity formed by the hydrophobic carbons of the glucopyranose monomers [45] where molecules can be housed, forming the so-called inclusion complexes, while the outer surface contains the hydroxyl moieties, constituting the hydrophilic part. Although natural cyclodextrins are hydrophilic, they have limited solubility in water and, thus, more water-soluble cyclodextrin derivatives are frequently used in formulations such as (2-hydroxy)propyl β-cyclodextrin (HPβCD). This is a derivative widely used to improve the solubility of hydrophobic drugs [46]. The increased solubility of Acz in Acz–HPβCD complexes (AczHP) has been investigated by other authors [47,48]. In addition, the use of these oligosaccharide complexes as topical permeation enhancers has been also studied [38,49]. Although these molecules, which are of high molecular weight and very hydrophilic, do not penetrate through the biological membranes, they ensure a high concentration of the dissolved drug (as a reservoir) on the membrane surface as reported in [50]. An interesting study suggested a multicomponent complex with HPβCD and triethanolamine as a promising approach to enhance the ocular bioavailability of Acz [51]. In ophthalmic formulations, many studies have demonstrated the tolerability and suitable toxicological profiles of HPβCD formulations [52,53].

As the IOP can be decreased by different pathways, the present work suggests, for the first time, a beneficial role of combining two different drugs with antiglaucomatous activity (TM and Acz) formulated as TM co-loading AczHP liposomes (CLLs) in reducing the IOP in an experimental model. It is well known that the ability of cyclodextrins to form inclusion complexes with hydrophobic molecules that are trapped in the aqueous liposome core [54] could potentially increase the drug/lipid mass ratio to levels higher than those achieved by conventional incorporation of drug in the lipid bilayer. The objective was to develop liposomes incorporating TM into the lipid bilayer and AczHP into the aqueous compartment. The combination of both drugs into a single formulation may offer a synergistic effect in glaucoma treatment, exploiting their different mechanisms of action over this disease. For this, a first screening stage concerning the optimization of liposomal formulations was carried out using Taguchi orthogonal arrays. Afterward, the effectiveness of AczHP co-loaded in TM liposomes in reducing the IOP of New Zealand rabbits’ eyes, compared to isolated drug in solutions, was evaluated.

## 2. Materials and Methods

### 2.1. Materials

L-α-phosphatidylcholine from egg yolk (EPC), cholesterol (Ch), didodecyldimethylammonium bromide (DDAB), and cholesteryl hemisuccinate (Chems) were provided by Sigma–Aldrich (Barcelone, Spain). Stearylamine (SA) was purchased from Fluka (Munich, Germany). Timolol maleate (TM), acetazolamide (Acz), and dodecylsulfate were purchased from Acofarma (Barcelone, Spain). (2-hydroxy)propyl-β-cyclodextrin (HPβCD, Eur. Pharm. 5th ed., D.S. 6.3) was supplied by Roquette (Lestrem, France). Acetonitrile (ACN), trichloromethane, methanol, ethanol, and 2-[4-(2-hydroxyethyl)-1-piperazinyl] ethanesulfonic acid (Hepes) were obtained from Panreac Química (Barcelone, Spain). Polycarbonate membranes (800 and 200 nm in pore size) were acquired from Millipore (Dublin, County Cork, Ireland). All other chemicals used were of analytical degree.

### 2.2. Quantification of Acetazolamide and Timolol Maleate

Reverse-phase high-performance liquid chromatography (HPLC) was used to simultaneously quantify Acz and TM. The assay was performed using a Hitachi Elite La Chrom system equipped with a reverse-phase column, an L-2130 isocratic pump, a diode array detector L-2455, and an L-2200 autosampler. As the stationary phase, a MerckLiChrospher 100 RP-18 (125 × 4 mm) column was applied.

The mobile phase consisted of sodium acetate trihydrate 0.05 M adjusted to pH 4.1 with glacial acetic acid as eluent A and ACN as eluent B. The gradient was 0–2.2 min: 90% A (10% B); 2.2–2.3 min: 90%→60% A (10%→40% B); 2.3–4 min: 60% A (40% B); 4–4.1 min: 60%→90% A (40%→10% B); 4.1–5 min: 90% A (10% B). The chromatographic separations were carried out at 25 °C, and the analysis was performed at a flow rate of 1.5 mL·min^−1^ (from 0 to 2.2 min) and 2 mL·min^−1^ (from 2.2 to 5 min). Twenty microliters of standard or sample solution were injected.

### 2.3. Phase Solubility Studies

Phase solubility studies on Acz with HPβCD were performed according to the method previously described by Higuchi and Connors [55]. An excess of marketed Acz (30 mg, approximately) was added to 20 mL of 0.01 M Hepes and 150 mM NaCl isotonic buffer solution at pH 7.4. The concentration range of the cyclodextrin solutions tested was from 0.01 to 0.1 M. Vials were kept in an orbital shaker (Unitronic^®^ OR, Selecta, Murten, Switzerland) at a constant temperature (25.0 ± 0.5 °C) for 72 h until complexation equilibrium was reached. Then, samples were filtered through 0.45 μm polyamide filters (Chromafil^®^ Xtra PA 45/25 Macherey, Nagel, Germany). The solubility of Acz in Hepes buffer solution was determined according to the same protocol without the presence of HPβCD. Acz concentrations in the filtrate were chromatographically calculated after suitable dilution (1:10 *v:v*) with the Hepes buffer solution.

A ternary system was obtained following the same procedure but adding an excess of Ch (20 mg). All the studies were realized in triplicate.

Following the same protocol we used in a previous paper for complexing curcumin with HPβCD and documented in the literature [56,57,58], the complexation constant (Kc) and the apparent stability constant (Ks) for AczHP complexes were obtained from the phase solubility diagrams by applying the equation:(1)$$\log\left(S_{t}-S_{0}\right)=\mathrm{nlog}\left[\mathrm{HP}\beta \mathrm{CD}\right]+\log\left(K_{c}S_{0}\right)$$
where S_t_ is the measured solubility in the presence of cyclodextrin solutions; n denotes the scope of the regression curve. The complexation constant (K_c_) was obtained from the intercept with the *y*-axis.

In the case of *n* next to the unit, it was assumed as a 1:1 ratio (A_L_ profile) and then K_s_ was calculated as follows:(2)$$K_{1:1}=\frac{\mathrm{slope}}{S_{0}\left(1-\mathrm{slope}\right)}$$
where the slope is the value found in the linear regression, and S_0_ is the aqueous solubility of the drug in the absence of HPβCD.

The enhanced solubility ratio (ESR) was calculated from the following equation:(3)$$\mathrm{ESR}=\frac{\mathrm{Acz}\;\mathrm{solubility}\;\mathrm{with}\;\mathrm{HP}\beta \mathrm{CD}}{\mathrm{Acz}\;\mathrm{solubility}\;\mathrm{without}\;\mathrm{HP}\beta \mathrm{CD}}$$

Likewise, the complexation efficiency (CE) of the Acz was calculated from the data on the phase solubility curve, using the expression [47]:(4)$$\mathrm{CE}=\frac{\mathrm{slope}}{1-\mathrm{slope}}$$

### 2.4. 1H-NMR

1H-NMR studies were performed on a Bruker Avance III spectrometer (Bruker BioSpin GmbH, Rheinstetten, Germany) operating at 300 MHz. Samples were prepared in D_2_O (Sigma–Aldrich, St. Louis, MI, USA) in concentrations of 2.8 mg·mL^−1^ for Acz and 17.2 mg·mL^−1^ for HPβCD. The spectra were acquired and referenced to the TMS at 0 ppm and adjusting the HDO signal to 4.710 ppm. Due to the low analyte concentration, all samples under analysis were subjected to HDO signal presaturation for better visualization of proton signals.

The two-dimensional ROESY (2D-ROESY) spectra were acquired on a Bruker AV-500 spectrometer (500 MHz) under the following conditions: 16,384 data points with 2048 increments, with 16 scans per increment, a mixing time of 250 ms, and a continuous wave spin block. Phase-sensitive data were collected using a water suppression scheme.

### 2.5. Method of Preparation

Liposomes were prepared through a thin-film hydration technique following the procedure previously detailed [59] with slight modifications. The total lipid amount was 0.1887 mmol, and the composition of the lipid bilayer was Chems (0.0812 mmol), EPC (0.0954 or 0.0867 mmol), SA or DDAB (0.0086 or 0.0173 mmol each one), and TM (0.0035 mmol).

Liposomal formulations consisted of unloaded liposomes (Empty-L), TM liposomes without AczHP (TM-L), AczHP liposomes without TM (AczHP-L), TM/AczHP co-loaded liposomes (CLLs), and Acz/TM/AczHP double-loaded liposomes (DLLs) containing Acz (0.0225 mmol) and TM in the bilayer and the Acz complexed into the aqueous core.

Briefly, all lipid components were dissolved in 8 mL of chloroform. The organic solution was transferred into a round-bottom flask. The solvent was evaporated under rotary evaporation (Büchi R-210 with Heating Bath Buchi B-491, Flawil, Switzerland), and the balloon was kept overnight in a desiccator to remove traces of organic solvents. The lipid film was hydrated and vortexed (VelpScientificaZx3, Usmate Velate, Italy) with 3 mL of AczHP (0.374 mmol HPβCD and 0.0675 mmol Acz) in Hepes buffer pH 7.4 for the AczHP-L, CLL, and DLL formulations. This step was realized at 58 °C for working above the phase transition temperature of lipids. Multilamellar vesicles (MLVs) obtained were quickly sealed in glass containers and stored in the dark at 4 °C until use.

With the aim of increasing the drug loading into the aqueous compartment of liposomes, the reduction in the vesicle laminarity was carried out by applying the extrusion or freeze–thawing (FAT) methodology. In this sense, MLVs were extruded in a Lipex Thermobarrel extruder (Northern Lipids Inc., Burnaby, BC, Canada) under airflow with nitrogen and moderate pressure of 600 psi. Eight hundred and 200 nm pore size polycarbonate membrane filters were used to obtain unilamellar liposomes (LUVs), each batch being passed five times through the extruder. The FAT technique involved subjecting the MLV dispersion to 10 freezing cycles for 30 s (in liquid nitrogen) and thawing in a bath at a temperature of 58 °C for 30 s. The breakage of vesicles with freezing and the subsequent formation of more homogeneous vesicular populations with thawing gives rise to more homogeneous samples of LUV [60,61].

### 2.6. Characterization of Liposomes

#### 2.6.1. Vesicle Size Analysis

Hydrodynamic diameters (d_H_) were measured by dynamic light scattering (DLS) using the Zetasizer Nano-S (Malvern Instruments, Malvern, UK). The d_H_ and the polydispersity index (PdI) were measured at room temperature for all preparations by diluting the liposomes 1:20 *v/v* in the same medium that they were made, Hepes buffer pH 7.4, avoiding any change in the vesicle integrity.

#### 2.6.2. Zeta Potential

The zeta potential (ZP) of the liposomes was determined by the electrophoretic light scattering technique after diluting the sample 20-fold in Milli-Q water. This parameter was calculated from electrophoretic mobility (μ) measurements using a Zetasizer Nano-S (Malvern Instruments, Malvern, UK) with a universal dip cell. This parameter leads to zeta potential (ZP) through the Smoluchowski equation: ZP = μηε, where η is the viscosity and ε is the permittivity of the solution.

#### 2.6.3. Morphology

Transmission electron microscopy (TEM) (Zeiss Libra 120) was used to visualize vesicle morphology. After diluting the sample (10 μL to 1 mL of Hepes solution), it was left to dry on a microscopic copper-coated grid (transmission electron microscopy grid support films of 300 mesh Cu). When the sample was dried, it was negatively stained by adding a drop of an aqueous solution of uranyl acetate (1% *w*/*v*), leaving for 10 min for reacting. Then, the excess solution was wiped with filter paper and washed with purified water twice. Finally, the grid was placed in the microscope, proceeding to view with an accelerating voltage of 75 kV at different magnifications.

#### 2.6.4. Drug Entrapment Efficiency (EE)

The percentage of TM and Acz entrapped into vesicles was obtained after discarding the unentrapped TM and Acz by centrifugation in a cooling centrifuge (Eppendorf Centrifuge 5804 R) for 60 min at 10,000 rpm, 4 °C [59]. The supernatant was filtered and analyzed by HPLC for drug content.

In addition, the pellet was resuspended with 0.5% *w/v* sodium dodecylsulfate aqueous solution for disrupting the vesicles. Samples were diluted, submitted to sonication for 10 min, and filtered for HPLC quantification [62].

Liposomal TM and Acz were expressed as the EE, calculated using the equation:(5)$$\mathrm{EE}=\frac{\mathrm{Amount}\;\mathrm{of}\;\mathrm{drug}\;\mathrm{quantified}}{\mathrm{Theoretical}\;\mathrm{amount}\;\mathrm{of}\;\mathrm{drug}}\;\cdot \;100$$

#### 2.6.5. ^31^P-NMR Analysis

For this assay, liposomal samples were prepared using the same procedure as above, but Hepes 2.85 mL and 0.15 mL D_2_O were used as the aqueous phase [63].

^31^P-NMR spectra were run on a Bruker AVANCE NEO 500 spectrometer at 202 MHz with proton broadband decoupling using a BBFO probe operating at 25 °C. The acquisition parameters were: 30° pulse 4.67 µs, relaxation delay of 2 s, 64 k data points, and 1500 scans. A 3.5 Hz exponential line broadening was applied before FT. Chemical shift values are quoted in parts per million (ppm).

### 2.7. Screening Step

To investigate the effect of the liposome composition on d_H_, ZP, and EE, a study was carried out using four factors (i.e., method for reducing size, cationic lipid, amount of cationic lipid, and double loading) assigned to two different levels: low and high (Table 1A). For this, an L8 Taguchi array was selected as the appropriate experimental design, with 8 experiments (Table 1B). All experiments were conducted in duplicate to estimate the variability of the data and, hence, the error values.

Analysis of variance (ANOVA) was used for statistical analysis of the results using software (DOE pack2000) with the aim of investigating which factors had remarkable results on the response parameters and the optimum conditions. Pareto diagrams were also used to select the conditions for minimizing or maximizing the responses evaluated.

### 2.8. In Vitro Release Studies

These studies were developed by following the dialysis method. In these studies, 1 mL of sample was placed in a dialysis bag (Spectra/Por^®^ 4, Rancho Dominguez, CA, USA, molecular cut-off 12–14 kD), previously rinsed and soaked for 1 h, sealing both borders with a dialysis clip. The bag was placed in 50 mL of artificial tears (NaHCO_3_ 0.2% *w*/*v*, NaCl 0.67% *w*/*v*, and CaCl_2_ 0.008% *w*/*v*), maintaining the stirring rate at 100 rpm and 37 °C (IKA^®^ RT10, Staufen, Germany); these conditions partially simulated the biological conditions [64,65]. Aliquots of dissolution medium were taken at predetermined time intervals (i.e., 15, 30, and 45 min and 1, 2, 3, 4, 5, and 24 h). Fresh medium replaced the volume extracted. The amount of drug dissolved with time was quantified using the HPLC method as previously described.

With the purpose of analyzing the release rate, the t_50%_ parameter was calculated, i.e., the time required for 50% of drug to be released. This assay was performed for CLL formulation and compared with identical formulations incorporating the drugs separately (TM-L and AczHP-L) and in standard solutions (TM-S and AczHP-S).

Absolute amounts of both drugs in the release study were: CLL (0.5 mg/mL TM and 5 mg/mL Acz), TM-L (0.5 mg/mL), AczHP-L (5 mg/mL), TM-S (0.5 mg/mL), and AczHP-S (5 mg/mL).

### 2.9. In Vitro Permeation Studies

The in vitro permeation studies on TM and Acz were evaluated using Franz-type diffusion cells. Experiments were planned for maintaining the sink conditions in the receiver compartment, considering the saturation concentration of Acz in Hepes buffer (0.98 mg/mL). All samples were adjusted to final Acz and TM concentrations of 5 and 0.5 mg/mL, respectively. Experiments were performed using Franz diffusion cells (diffusion area of 3.14 cm^2^). Hydrophilic polysulfone membranes (Tuffryn^®^; Pall Corporation, Port Washington, NY, USA) were impregnated with a gelatin solution (12% *w*/*v*) in order to mimic the high protein content of the stroma (mainly collagen). It is well known that gelatin results from the partial hydrolysis of collagen, and it can generate a flexible gelled film [66]. The modified membrane was then impregnated with Hepes buffer solution (pH 7.4) for 30 min with the aim of saturating the membrane with the aqueous medium used in the receiver compartment. The preparation procedure for the Franz cells has been described previously by the authors of [28]. The percentage of Acz diffused did not rise to 100%, even after 24 h of diffusion and in respect to the sink conditions. Aliquots of 1 mL were collected after 0.25, 0.5, 1, 2, 3, 4, 5, and 24 h according to international guidelines, being the same volume replaced with fresh medium kept at the same temperature. The samples were quantified by HPLC. Permeation tests on CLL, TM-L, AczHP-L, TM-S, and AczHP-S were performed.

Acz and TM permeability parameters were calculated by plotting the amount of permeated drug across the membrane (mg/cm^2^) as a function of time (min), using the linear segment of the curves (initial phase). The steady-state flow, J, was determined with the following equation [67]:(6)$$J=\frac{\delta Q}{A\delta t}$$
where Q indicates the amount of drug impregnated, A is the diffusion area, and t is the exposure time. The permeability coefficient, P, in each case was calculated using the following equation:(7)$$P=\frac{J}{C_{0}}$$
where C_0_ is the concentration of drug in the donor medium.

### 2.10. Stability Studies

The physical stability of the vesicles and their ability to retain the drug were assessed for 3 months at 2–8 °C. Samples were withdrawn periodically and analyzed for EE, d_H_, PdI, and ZP.

### 2.11. In Vivo Hypotensive Efficacy Studies—IOP Determination

The experiment was performed using normotensive New Zealand male rabbits 2–2.5 kg in weight, and each rabbit was kept in an individual cage, allowed free access to water, fed a standard diet, and maintained in a controlled 12/12 h light/dark cycle. The handling and care of animals were based as established by the Association for Research in Vision and Ophthalmology (ARVO) and the European Communities Council Directive (86/609/EEC). This study procedure was approved by the Institutional for the Care and Use of Experimental Animals (CICUAL) of the School of Medicine Sciences, National University of Cordoba, Argentina (Res. 44/17). All efforts were made to reduce the number of animals used.

IOP was measured with a TONOVETs rebound tonometer (Tiolat, Helsinki, Finland), for which topical anesthesia was not required. For each eye, IOP was set at 100% with two basal readings taken 30 min before and immediately after the instillation.

The formulations under study were applied in 10 normotensive rabbits, and the control group was evaluated in 5 of them. Ophthalmic preparations were instilled in a single dose in the lower eyelids of the rabbit using a micropipette (50 μL). It was applied to both eyes. The IOP determinations were performed once every hour over the following 7 h. For control purposes, rabbits received the formulations without the hypotensive agent (Empty-L). The administration protocol included at least a 48 h washout period between experiments. The experiments were performed in triplicate.

The effect of the TM and Acz formulations described in these in vivo studies were compared using the maximum hypotensive effect of the drug (%), the area under the ΔIOP curve (%) as a function of time (h) from 0 to 7 h (AUC), and the average time (h) in which the duration of the hypotensive effect was maintained.

### 2.12. Statistical Analysis

Statistical differences between two mean values were evaluated by a two-tailed Student’s *t*-test, and an analysis of variance (ANOVA) was applied when necessary. The results obtained were taken as significantly different at *p*-values < 0.05. The reduction in intraocular pressure is expressed as the means of 6 tests and the standard error of the mean (SEM).

## 3. Results

### 3.1. Phase Solubility Diagrams

In this study, HPβCD was used to increase the solubility of Acz (0.98 mg/mL in water, 25 °C). Based on previous results with liposomes and in literature related to instability when a complexing agent was incorporated into the formulation, we analyzed the influence that the inclusion complex AczHP may have had on some components of the liposomal formulation such as Ch.

The phase solubility diagram of the binary system, Acz/HPβCD (Figure 1a), evidenced that the solubility of the drug increased linearly as the HPβCD concentration increased. Acz solubilities decreased when Ch was added in the assay, having a straight profile slope of 0.109 versus 0.160 in the case of Acz/HPβCD without Ch. However, the complexation ability of Chems by HPβCD was determined following the same procedure, and only a slight slope less than Acz/HPβCD was obtained (0.155).

The complexation constant (K_c_) and *n* were calculated from the linear fit of the curve according to Equation 1 (Figure 1b). The results obtained are reported in Table 2. Values of the slope *n* suggest the formation of a 1:1 soluble inclusion complex according to the model proposed by Higuchi and Connors [55], obtaining A_L_-type diagrams. The presence of Chems in the ternary complex may be favorable to the complexation process, compared with the Acz/HPβCD or Acz/HPβCD/Ch mixtures, as the K_c_ values indicated.

On the other hand, the same equation was used to determine the value of the formation constant of the Acz/HPβCD complex. The Acz stability constant (K_s_) in the HPβCD aqueous solution at 25 °C was 54.8 M^−1^ in the binary complex Acz/HPβCD (Table 2). This value remained slightly higher when Chems was added (62.5 M^−1^); however, a lower K_s_ was obtained after adding Ch, suggesting a loss of stability in the complex.

The presence of HPβCD quintupled the solubility of Acz in aqueous solution (which is theoretically 0.98 mg/mL at 25 °C) as demonstrated by the ESR ratio in Table 2. Moreover, this parameter improved in the presence of a third compound, Chems.

In addition, the complexation efficiency (CE) was evaluated to determine the affinity of these molecules in the complexation process. The values obtained for the Acz/HPβCD and Acz/HPβCD/Chems complexes (0.191 and 0.183) mean that approximately one out of five cyclodextrin molecules in solution formed a water-soluble complex with Acz; however, in the case of Acz/HPβCD/Ch (0.123), one out of ten HPβCD molecules in solution formed the complex with Acz [47,68].

All experimental data for the phase solubility diagrams are collected in Appendix A.

### 3.2. 1H-NMR

In order to explore the possible inclusion mode of the Acz/CD complex, we compared the 1H-NMR spectra of Acz in the presence of the host HPβCDs (Figure 2). Figure 2a shows the proton allocation of Acz, while Figure 2b shows the 1H-NMR spectra corresponding to HPβCD, Acz/HPβCD (1:1), and Acz. Chemical shifts in the Acz and CD protons in both the free and complexed forms are summarized in Table 3.

Due to the structure of Acz, the protons that could be recorded were those corresponding to the acetamide group, which showed a singlet signal at 2.253 ppm with respect to the reference (TMS).

To gain more conformational information, we obtained 2D ROESY of the inclusion complexes of Acz with HPβCD. The ROESY spectrum showed appreciable correlation of the H-3 proton of HPβCD with the CH_3_ of Acz.

### 3.3. Screening Step

With the aim of fixing the composition and disposition of the lipid bilayer when Acz–HPβCD (AczHP) was loaded into the aqueous core, a screening study was performed using the design of experiments. The formulations were analyzed in terms of d_H_, PdI, ZP, and EE. As shown in Figure 3a, the results followed the same tendency for d_H_ and PdI. The d_H_ obtained ranged from 607 ± 150 to 1551 ± 45 nm and PdI from 0.1 ± 0.1 to 1 ± 0. The smallest average vesicle size was obtained in trial 3 in which DDAB was used as the charge donor agent in the amount of 0.0086 mmol, and extrusion was carried out on the double-loaded samples.

Values of ZP ranged from 13.15 to 24.25 mV, and a relationship between the amount of charge donor agent in the design and the cationic ZP values was observed as shown in Figure 3b.

Concerning the EE, trial 5 showed the highest value, having 0.0086 mmol of SA and submitting non-double-loaded samples to FAT (Figure 3c).

The influence of factors on the characterization parameters was statistically evaluated to select their significance on the analyzed responses. ANOVA data (Table 4) show the factors with a higher contribution percentage of significant differences on the responses evaluated as follows: F1, F2, and F4 on d_H_; F1 and F2 on PdI; F1 and F3 on ZP; F1 and F2 on EE Acz. This means that we worked on these parameters for optimizing the liposomal formulation.

It is important to clarify that the Taguchi design includes the analysis of interactions among the main factors. This is why the PdI and EE responses showed a sum of percentages of contributions of the main factors less than 100%, corresponding the rest to the interactions among factors. Since some two-factor interactions in the Taguchi design were confounded with some other main factors, it was not possible to distinguish which of them actually possessed statistical significance.

With the aim of elucidating the signing sense of the effects on the responses studied, Pareto charts were acquired. Figure 4a (d_H_) shows the significant positive effect for F1 (on the right side) and the significant negative effects for F2 and F4 (on the left side). As the objective was to minimize the vesicle size to improve the permeation behavior at the corneal level, the lower level of the positive effects and the higher level of the negative effects were thus selected; in this case, extrusion, DDAB, and double loading were chosen as significant effects. The same procedure was followed for designating the parameters for the PdI (minimizing), ZP (maximizing), and EE Acz (maximizing) responses.

These factors and their levels were also fixed for the PdI analysis. In addition, the use of 0.0086 mmol DDAB was favorable for minimizing this parameter.

Concerning ZP, this parameter was maximized using FAT as the method for obtaining LUVs, non-double-loading, and 0.0173 mmol of the charged agent. This last effect was the most significant, with a contribution degree of 87%.

Finally, liposomes submitted to FAT and containing SA as the charged agent showed higher values of EE.

The summary of the extracted information from this analysis is presented in Table 5.

Thus, the composition of the optimized liposome formulation (CLL) included AczHP in the aqueous phase. The quantitative optimized composition in the lipid bilayer was EPC 0.0867 mmol, Chems 0.0812 mmol, DDAB 0.0173 mmol, and TM 0.0035 mmol. HPβCD 0.3 mmol and Acz 0.0589 mmol were included in the aqueous phase (Hepes 3 mL). In addition, two control liposomal formulations were prepared with TM (TM-L) and AczHP (AczHP-L), separately.

The optimized sample in the composition was finally extruded by 200 nm pore size to enhance the permeation mechanism of the vesicles without modifying their physicochemical properties.

### 3.4. Characterization of the Optimized Formulation

Characterization values of the optimized sample (CLL) as well as the control formulations (TM-L and AczHP-L) and unloaded (Empty-L) are shown in Table 6.

Size and PdI values were according to the expected results, having sizes next to the pore size used in the extrusion and homogeneity of size distribution. d_H_ was higher after entrapping AczHP in the aqueous core (CLL and AczHP-L), as is logical. ZP-acquired values ranged from 18.9 in AczHP-L to 32.3 mV in TM-L. This difference was lower when both drugs were enclosed in the same formulation (CLL). On the other hand, higher EE*s* for Acz and TM were obtained in the optimized batch (CLL).

Figure 5 shows TEM images of Empty-L (a), TM-L (b), AczHP-L (c), and CLL (d). All liposome formulations showed nano-sized globular shapes, and their sizes were lower than those measured by DLS, probably due to the loss of water for analysis.

Results of the ^31^PNMR analysis are shown in Figure 6. For Empty-L, the resonance profile denotes a distinguishable broadband around 0 ppm. For TM-L, the intensity of the peak increased substantially compared to the peak of liposomes without drug, with a narrower band observed; in addition, a low signal on the left side was also obtained. For AczHP-L, the spectrum signal was lower compared to liposomes containing TM. Finally, the CLL sample containing both drugs showed a spectrum similar to that of AczHP-L.

### 3.5. In Vitro Release and Permeation Studies

In vitro release studies were performed in artificial tears at 37 °C using the optimized liposome formulation (CLL) and comparing it to the control solutions of both drugs (i.e., AczHP-S and TM-S) and liposomal formulations containing the single drugs (AczHP-L and TM-L).

The in vitro release profiles (Figure 7) revealed the influence of the liposomal encapsulation of both drugs. For AczHP-S and TM-S, the drugs were rapidly released through dialysis membranes, and more than 80% of the drug amount was recovered in the dissolution medium after 24 h, while for the liposomal formulations, only the CLL batch could reach this value. All formulations exhibited rapid release of AczHP during the first hour (first phase) according to the free AczHP release, which was followed by a prolonged release for 24 h (second phase) (Figure 7a). TM demonstrated more similar release patterns than AczHP, but more than 60% of TM was released within 2 h from all formulations (Figure 7b). The t_50%_ values (Table 7) denoted a delayed release effect in the AczHP samples from liposomes compared with TM, having higher t_50%_ values overall in AczHP-L. During the first hour, the amount of drug released from liposomes was practically linear over time, attributable to a burst effect of the free drug unentrapped, following the same tendency as AczHP-S. Although this effect was common for the AczHP and TM samples, an enhanced Acz release process was appreciated in CLL compared with AczHP-L. For the TM samples, the amount released into the lipid bilayer was delivered more quickly into the dissolution medium, even improving when AczHP was co-loaded into the aqueous core of the liposomes.

In vitro permeation studies were carried out using Franz diffusion cells, and samples were placed into the donor compartment with the aim of analyzing the amount of drug permeated over 24 h through an artificial membrane. The permeation rate was also evaluated.

Table 7 shows the permeation parameters of TM and AczHP from the different formulations. Firstly, the sustained release effect of the liposomes was patent, as observed in the flux rate values, being lower in the liposomal formulations than in the standard solutions. The results also reveal that the amount of AczHP permeated was significantly higher than TM in all formulations; this is logical considering the initial loading of both drugs. However, for the same drug, the amount of drug permeated was independent of the formulation. The same result was obtained for the flux rate, but significant differences comparing the liposomal formulations (CLL (Acz) and AczHP-L) with AczHP-S were obtained for this parameter, being lower for the liposome samples. After analyzing the permeability coefficient, the delayed effect of AczHP for all liposomal samples in the permeation mechanism compared with the TM formulations, was evidenced.

### 3.6. Stability Studies

The stability studies on the CLL formulation were carried out over 3 months. Vesicle size (d_H_) for all extruded formulations decreased by approximately 5 nm, and ZP varied approximately 2 mV over the storage period. These results are according to the decreased values of 31 nm presented in previous studies [28]. TM and Acz were found to be stable when analyzed in terms of vesicle size, PdI, ZP, and EE at 4 °C (Appendix A).

### 3.7. Hypotensive Efficacy of the CLL Formulation

Independent experiments were performed on 10 rabbits (*n* = 10) to evaluate the hypotensive effect of each formulation. Two standard solutions of TM and Acz, separately, with concentrations of 0.5 mg/mL (TM-S) and 0.5 mg/mL (Acz-S) or 5 mg/mL (AczHP-S), respectively, were used as the reference.

In vivo studies conducted with the liposome formulation without HPβCD (0.5 mg/mL Acz), named Acz/TM-L, did not show an improvement in the reduction of IOP compared with the control solutions; however, this slight decrease in the IOP remained throughout the experimental trial compared to the control solutions (Figure 8a). This affirmation was demonstrated from the AUC in Acz-S (24.44 ± 6.67%·h) and TM-S (38.10 ± 7.65%·h), which were slightly smaller than the Acz/TM-L formulation (54.32 ± 10.13%·h) as depicted in Figure 8c.

The addition of HPβCD to improve the solubility of Acz allowed for the incorporation of ten times more of the drug, significantly improving the therapeutic efficacy of the formulation as shown in Figure 8b.

Data collected in Figure 8d reveal that the maximum hypotensive effect of CLL (37.29 ± 2.86%) was very significant based on the IOP reduction of TM-S (11.58 ± 2.58%) and AczHP-S (26.77 ± 2.36%). Currently, the concentrations of TM and Acz used commercially are 5 mg/mL and 250 mg every 8 h, respectively. In vivo studies conducted after topical application in rabbits showed that the hypotensive effect of TM (0.5 mg/mL) and Acz (5 mg/mL) from the CLL formulation markedly increased. In addition, the residence time of the hypotensive effect was greater in the CLL formulation with respect to the control solutions (i.e., AczHP-S and TM-S). Therefore, the effectiveness of the formulation was superior to the standard solutions and the liposome formulation without HPβCD. As Figure 8d shows, the AUCs for TM-S, Acz-S, AczHP-S, and Acz/TM-L were significantly lower than the values obtained with CLL (126.89 ± 9.96%·h).

## 4. Discussion

The low solubility and poor bioavailability of Acz are the main drawbacks that limit its use via the topical route. In this study, the encapsulation of HPβCD, including the complex, into TM-loaded liposomes was conducted in order to offer an alternative glaucoma treatment based on the synergistic effect of both drugs in reducing the IOP.

For the Acz/HPβCD binary system, the aqueous solubility of Acz was found to increase linearly as a function of the concentration of HPβCD. According to the Higuchi and Connors classification, the phase solubility diagrams of the Acz/HPβCD complexes were of the AL-type (*r*^2^ = 0.9978), indicating the formation of soluble complexes [69]. From the solubility diagrams, the extent of complexation was calculated by the apparent 1:1 stability constant, K_s_. This value (54.8 ± 0.9 M^−1^) indicated that the complex formed was poorly stable. However, the assessment of the Acz/HPβCD complex by 1H-NMR clearly demonstrated the presence of the framework protons of the Acz molecule, consistent with significant solubilization. As is well known, changes in the NMR signals of the protons located in the inner CD cavity (H3, H5 and, to a lesser extent, H6) were linked with the entrance of a guest molecule in the CD cavity. Despite the reported chemical shift changes for H3 and acetamide protons, all the chemical shift variations from the NMR study were of low magnitude, indicative of a weak host–guest interaction, in consonance with the polarity data for the Acz (logP = −0.26) [70,71]. In order to obtain further data on the formation of a true inclusion complex in solution as well as additional information on the geometry of the complex, H-H ROESY two-dimensional NMR studies were performed (Figure 2c). As noted above, the most remarkable fact was the presence a weak cross-signal between the internal H3 protons of the CD and the Acz acetamide protons. These findings confirmed the interaction between Acz and HPβCD in aqueous medium. As with the modifications of the chemical shifts from the NMR studies, the weakness of the ROE signal is in agreement with the low stability constant value reported from the phase solubility studies. These results are in concordance with others previously reported in the literature [51]. Lastly, the above data would suggest that the portion of the Acz molecule was not completely oriented towards the CD cavity or even the coexistence of different modes of complexation between Acz and HPβCD. These hypotheses have yet to be confirmed by additional data and exceed the objectives of the present work.

In an attempt to evaluate the effect of CDs on the complexing capacity of liposomal components and reviewing the possibility that Ch, a lipid component in the lipid bilayer of liposomes, may be partially complexed by HPβCD, as reported in the literature [72,73], the behavior of them incorporated in a simultaneous form was analyzed. The solubility phase diagram showed that the addition of HPβCD in the Ch-containing formulation was unfavorable due to the affinity of CDs by this steroid. The phase solubility studies revealed a linear relationship in the aqueous drug solubility with an increase in HPβCD concentration, but a lower slope was obtained compared to the Acz/HPβCD system (0.1094 ± 0.0041 vs. 0.1601 ± 0.0004), which implies less Acz solubilized, while HPβCD increased. Other complexation parameters, such as Ks (47.9 ± 1.8 M^−1^) and others that will be discussed, indicated that the presence of Ch was unfavorable to the complex’s stability. To verify that the non-solubilized fraction of Acz was due to the presence of Ch, this steroid was replaced by a more soluble derivative, Chems, which has a different polarity, and there was no evidence in the literature that it interacts with the CD. Chems is a dicarboxylic acid monoester resulting from the condensation of the hydroxyl group of Ch with one of the carboxy groups of succinic acid. This surfactant structure avoids complexation. Similar phase solubility diagrams in the presence of Chems (similar slope to Acz/HPβCD) confirmed that the presence of Ch in the formulation was the cause of the lower Acz solubility. There was a drug shift from the hydrophobic cavity of the CD when Ch was incorporated as a ternary component. Comparison of Kc (M^−1^) and Ks (M^−1^) in both systems (50.3 ± 2.3 vs. 54.8 ± 0.9 in Acz/HPβCD and 41.1 ± 7.5 vs. 47.9 ± 1.8 in Acz/HPβCD/Ch) revealed that other complexes than 1:1 are possible. These results are in agreement with those of other authors, who concluded that Ch and HPβCD give rise to complexes with a stoichiometry of Ch·2CD [74]. Other studies were conducted to modify the content of this steroid in cell membranes by incubating cells or model membranes with CD, confirming that these sugar derivatives can be used to extract Ch from cell membranes by forming inclusion complexes [75]. This may lead to membrane disruption and the decreased solubilization of the drug as shown in Figure 1.

Comparative analysis of the three systems analyzed emphasized that according to Loftsson and Brewster [76], the Ks of the three complexes, all with CD, belonged in the optimal range for good bioavailability. However, comparing their values, it was concluded that HPβCD/Acz/Chems exhibited parameters more favorable for enhancing Acz solubilization (ESR 6.4) and complex stability (Ks 62.5 M^−1^) than HPβCD/Acz/Ch. Most probably, the addition of hemisuccinate groups to βCD strengthened the interaction with drug molecules and weakened the interactions among CD molecules, leading to higher stability and better solubilization of Acz. These results agree with those of other authors [77]. On the other hand, the decrease in the S_0_ value of Acz in the presence of Ch or Chems in the aqueous medium compared to Acz alone opens the possibility of having some mechanism of drug retention in the steroid structure, such as micelles, as has been reported [78]. These results are in concordance with the CE values, giving CE in the order HPβCD/Acz > HPβCD/Acz/Chems > HPβCD/Acz/Ch, which demonstrates that with Chems, HPβCD can complex better with the guest Acz than with Ch. The solubilizing properties of this ternary system for Acz may be an interesting property. Therefore, Chems was selected as the steroid bilayer–rigidity agent for further studies.

In the current study, a DoE approach was used to investigate four factors (two composition factors and two methodological factors) in a liposomal system with respect to their physicochemical properties and Acz encapsulated (%). DoE approaches are increasingly used for the formulation and process development of liposomes [79,80,81]. In this study, it was found that the cationic lipid (SA or DDAB) and its concentration affected liposomal characteristics such as the size, PdI, ZP, and EE of Acz. The inclusion of an additional procedure for reducing the size also affected these parameters, mainly those related to size (d_H_ and PdI). This result was important for the further reduction in the sample size for obtaining CLL, and this conclusion has been obtained in other studies in which the extrusion was incorporated in the final step of liposome production to homogenize the vesicle size [82,83,84]. On the other hand, it was noticed that the amount of the charged agent was important, too, and had a significant effect on ZP, having a percentage of contribution of 83% (Table 4). Pareto charts indicated that the higher level (0.0173 mmol) provided the more positive values; thus, ZP was maximized. In these conditions, DDAB was selected as the cationic lipid for liposome formulation. This surfactant has demonstrated advantageous properties for liposomal formulations in terms of reduced size and lower PdI than SA. DDAB is a double-chain amphiphilic lipid positively charged in aqueous solutions (Figure 3d), while SA is an alkylamine positively charged at physiological pH (pKa 10.3) with a saturated chain that provides rigidity to the bilayer structure (Figure 3d) [85]. These structural differences mainly affect the size by providing greater flexibility and higher capacity for packing the bilayer as has been demonstrated in other studies [86]. On the other hand, and taking in consideration the low complexation strength of the Acz/HPβCD complex, a possible interaction between CD and SA can destabilize the lipid bilayer, resulting in larger sizes as was postulated by Jara et al. (2004) [87].

Acz was efficiently encapsulated into liposomes as AczHP, reaching an EE of 72%, compared to previous values of un-complexed Acz in other studies (22%). As discussed previously, the Acz/HPβCD complex enhances the solubility of Acz in aqueous solution based on their complexation parameters [87].

Lipid composition is considered to have significant effect on the EE of Acz in the sense that rigid structures provide more lipid barriers to escape the drug outside the liposome. For this reason, liposomal formulations with SA exhibited a higher EE of Acz than DDAB. Studies realized by other authors confirm the formation of a “compact multilayer” after freezing–thawing that favors the retention capacity of molecules [88]. The surfactant properties of DDAB decrease the interfacial tension with the external medium, favoring the exit of Acz into the dissolution medium [86]. In addition to solubility, the presence of TM in the lipid bilayer may favor Acz release from the inner compartment of vesicles.

The optimized formulation was obtained considering the degree of the statistical contribution of the factors on the evaluated responses as shown in Table 4. Therefore, the lipid bilayer was composed of EPC, Chems, 0.0173 mmol of DDAB as cationic lipid, and TM, whereas in the aqueous core, AczHP was added. The characterization parameters of the optimized formulation after extrusion were favorable for ophthalmic administration. In addition, simultaneous encapsulation of both drugs showed a slight increase in the EE of both TM and Acz, compared with other studies in which a lower EE was achieved, probably due to the steric competence by the same mechanism [89].

In an attempt to characterize the conformational state of drugs in the hydrophobic area of the liposomes, the interactions of the drug molecules (TM) with the polar groups of the lipids, and changes in the structure in the presence of AczHP, a ^31^PNMR technique was applied. Phosphatidylcholines (PCs) are the most prevalent phospholipids used in the manufacture of liposomes and are frequently used as a model mimicking the properties of the lipid membrane [90]. The PC molecule has a phosphate group; thus, ^31^PNMR spectroscopy represents a feasible technique to access the electronic environment of the ^31^P atom. The ^31^PNMR spectrum of the PC was centered around 0 ppm and was highly dependent on the conformation, orientation, and dynamics of the phospholipid molecules. The results obtained on the conformational changes in the membrane bilayer caused by the presence of TM, Acz, and charge-inducing substances provided interesting data to explain the interactions between them.

As Figure 6 shows, Empty-L denoted a distinguishable isotropic peak around 0 ppm and a shoulder. As confirmed by other authors, PC lipids in water spontaneously form a bilayer structure that gives rise to an asymmetric line shape of their ^31^PNMR spectra, with a low field shoulder and a high field peak [90]. For TM-L, the spectrum had a peak of greater intensity with respect to the reference peak. According to the literature, this narrower signal could correspond to micelles [91], agreeing with the size results obtained (Figure 5b). However, this signal can also be produced due to the rearrangement of the P molecule in the dissolution medium when the magnetic field is applied to it; the location of the TM in the liposome bilayer could induce, by repulsion, a possible interaction between DDAB and PC. The positive charge of DDAB can interact with the negative charges of PC causing a slight shift in the peak. Previous studies by a research group [63] reported this interaction between the cationic charged agent and the negative charge of the phospholipid. Obviously, the isotropic peak slightly away from 0 ppm was related to a difference in the ^31^P resonance because of a different mobility of the phosphate group in these systems. In fact, alterations in the mobility of the negative-ionized phosphate group of PC can be expected in the case of interactions with the positive-ionized amino groups of molecules such as chitosan [91]. Ionized TM in the bilayer could be the reason for obtaining more positive values of ZP.

AczHP-L in Figure 6 showed a ^31^PNMR spectrum without a displacement of the peak compared to TM-L, returning to the initial peak type (Empty-L). The presence of AczHP in the aqueous space did not affect the bilayer’s integrity. As the volume of the aqueous core increased, the curvature of the bilayer was modified, producing a rearrangement in the bilayer where the PC was oriented towards the outside. These results agree with the less positive ZP values in AczHP-L (Table 6).

Finally, the similar spectrum obtained for CLL and that of AczHP-L can be attributed to the stabilizing effect of AczHP in the formulation, since in the absence of it, the TM/DDAB destabilizes the bilayer by interacting with the phosphate groups of the PC. From this result and together with the physicochemical characterization, it seems that CLL was the most stable formulation among the others evaluated.

Drug release from liposomes is one of the most important criterions to be considered in drug delivery. Viewing the in vitro release profiles, these drug behaviors by the formulations have been previously obtained by other authors [77]. As expected, AczHP-L delayed the Acz release compared to AczHP-S in which the release of Acz was more pronounced [92,93]. The same behavior was observed with the CLL batch, especially in the first stage, in which the release rate of Acz was more noticeable in the CLL formulation than AczHP-L. The presence of TM in co-loaded liposomes may be the cause of this higher Acz release in the first phase due to the release of TM that is probably incorporated in the external monolayer of the membrane. Moreover, the release rate was slower in the second phase in the AczHP-L formulations compared to AczHP-S, because complexed Acz in the aqueous compartment of the liposomes must overcome more barriers to release from the system than TM [54]. The whole release behavior of both drugs could be explained by the first release of TM due to the fact of its high solubility in aqueous medium, which could favor vesicle destructuring, hindering the release of Acz. Our results agree with previous research comparing the sustained release of drugs in cyclodextrin in co-loaded liposomes compared to drugs in cyclodextrin in conventional liposomes [94].

The in vitro permeation results could be explained by the fact that a part of the sample remained in the donor chamber due to the hydrophilic nature of the membrane (data not shown). In the case of the TM formulations, almost 100% drug permeation was achieved. AczHP-L gave rise to a lower flux rate (J) and permeability constant (P) compared to AczHP-S (*p* < 0.01). However, TM only offered statistical significance between the TM-S and CLL formulations (*p* < 0.01) for the P parameter. The lipophilicity and the location of the drug in the liposome are considered as significant factors influencing the membrane permeability of these drugs [95]. Comparing both drugs for the same formulation, the influence of the drug concentration on the donor compartment was evident. The higher concentration gradient in the case of Acz caused an improvement in the permeation parameters as reported in Table 7. In our study, the presence of TM in CLL showed a slight increase in the permeation of AczHP compared to AczHP-L. This enhancer effect of TM in the lipid bilayer was exhibited in the in vitro release profiles, showing the same relationship as the permeation behavior. Therefore, it can be concluded from the study that the optimized formulation could act as an enhancer of permeation for Acz and TM because of the presence of a lipophilic structure, cationic charge, and HPβCD, according to other authors [96].

In vivo studies were realized in rabbits to evaluate the hypotensive effect of the formulations tested in vitro. First, in vivo studies were performed with the liposome formulation without incorporating HPβCD to analyze the effect of the combined formulation in the absence of cyclodextrin and at 0.5 mg/mL Acz. Afterwards, in vivo studies were developed for the CLL formulation. For this, TM-S (0.5 mg/mL), Acz-S (0.5 mg/mL), and AczHP-S (5 mg/mL) were used as the control samples.

As was postulated by other authors, the results of physicochemical properties obtained in our formulations are encouraging since they could become a key option for drug delivery through sclera and conjunctiva to reach the target site (ciliary body) more efficiently, with greater bioavailability [97,98].

In addition, the incorporation of CDs to improve the solubility of Acz allowed for the incorporation of this drug, searching for a synergistic effect with TM. Therefore, the success of the incorporation of this molecule into the formulation was corroborated, significantly improving the therapeutic efficacy of the formulation since, without it, no improvement was observed in the reduction of IOP with respect to the control solution.

The IOP (%) profiles in Figure 8a,b revealed that the hypotensive effect of the liposomal formulations was concentration related, and it was improved in the presence of HPβCD. The positive effect of liposomes and the use of cyclodextrins are in agreement with other studies [46,99]. The enhancement of drug permeation through the corneal membrane by liposomes was attributed to different mechanisms: the similarity of the liposomal composition to the biological membranes, which facilitates the interaction and/or fusion between them, and the interaction of phospholipids and charged lipids with the cell membrane, perturbing and disrupting its integrity. The observed enhancement of the IOP reduction in the presence of HPβCD could be attributed to solubilizing and partitioning effects over Acz [100,101,102].

In addition, the formulation proposed here maintained the hypotensive effect of TM and Acz, reaching a 37.29 ± 2.86% decrease in IOP. This value is very significant based on the IOP reduction of the TM-S (11.58 ± 2.58%) and AczHP-S (26.77 ± 2.36%). This can be attributed to the mimicking effect of liposomes on the lipid composition of the pre-ocular tear film, being readily internalized by corneal cells. On the other hand, cationic liposomes can prolong the drug residence time in the precorneal area at the absorption site.

Therefore, the resulting formulation could be used as a promising nanosystem for therapy of patients with elevated IOP. TM and AczHP co-loaded vesicles showed a significant reduction in IOP due to the contribution of TM in reducing aqueous humor production and the key role of Acz by inhibiting carbonic anhydrase. Therefore, the combination of drugs and their effect through dual mechanisms of action restore physiological IOP levels for the effective treatment of glaucoma.

## 5. Conclusions

The results of the inclusion complexation behavior, characterization, and binding ability of Acz with HPβCD alone and in the presence of Ch or Chems showed that HPβCD could enhance the water solubility of Acz, despite the binding ability of the complex. Ch disturbed the stability and solubility parameters of Acz due to the fact of its competence by CD; thus, Chems was selected for further liposome formulation studies.

AczHP and TM were successfully encapsulated into liposomes by employing the strategy of co-loading. The use of DoE for screening the effect of several formulations and methodological variables was useful for reducing the number of experiments. Optimizing the composition of the lipid bilayer and the method for reducing the vesicle size was crucial to improving the physico-chemical properties and encapsulation efficiency of both drugs. Moreover, in vitro release and permeation studies demonstrated that the co-loaded formulation extended the drug release more effectively than control solutions, also showing an improvement in drug release compared to the liposomal formulations with a single drug. This nanocarrier demonstrated high effectiveness in reducing and prolonging the IOP, suggesting that the synergistic effect of TM and Acz on aqueous humor retention, the permeation enhancing ability of HPβCD, and the role of liposomes as permeation enhancers and reservoirs are responsible for the success of this co-loading strategy for glaucoma therapy.

## Figures and Tables

**Figure 1 pharmaceutics-13-02010-f001:**
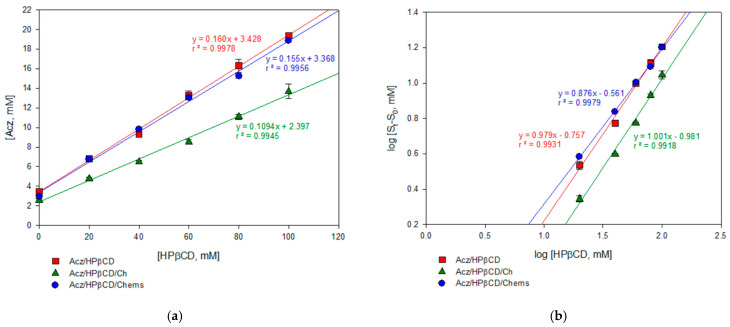
Solubility phase profiles of acetazolamide–(2-hydroxy)propyl-β-cyclodextrin alone (Acz/HPβCD) and in the presence of cholesterol (Acz/HPβCD/Ch) or cholesteryl hemisuccinate (Acz/HPβCD/Chems): (**a**) concentration of Acz solubilized vs. HPβCD concentration; (**b**) log–log plotting to calculate Kc.

**Figure 2 pharmaceutics-13-02010-f002:**
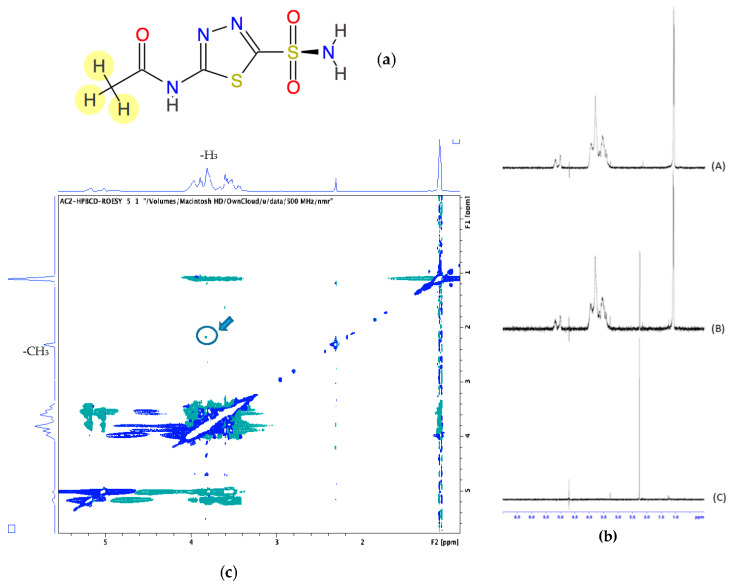
(**a**) Chemical structure of acetazolamide (Acz); (**b**) 1H-NMR corresponding to HPβCD (**A**), Acz/HPβCD (1:1) (**B**), and Acz (**C**). (**c**) 2D-ROESY spectrum of Acz/HPβCD (1:1). The highlighted circle indicates the crosspeak between H3 and the protons of the acetamide group of Acz.

**Figure 3 pharmaceutics-13-02010-f003:**
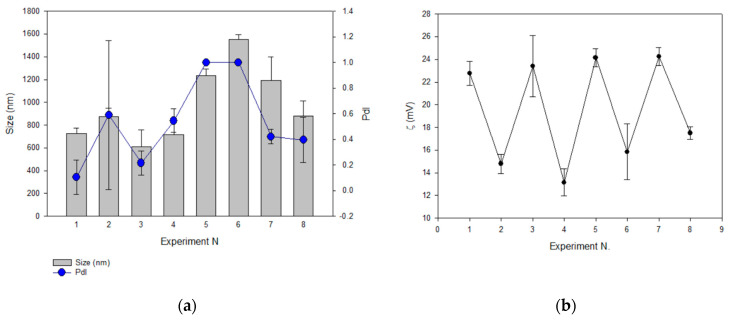
The effect of formulation parameters on (**a**) vesicle size (d_H_, nm) and PdI; (**b**) zeta potential (ZP, mV); (**c**) encapsulation efficiency of Acz (EE). Experiments were conducted in duplicate. (**d**) Chemical structures of DDAB (**top**) and SA (**bottom**).

**Figure 4 pharmaceutics-13-02010-f004:**
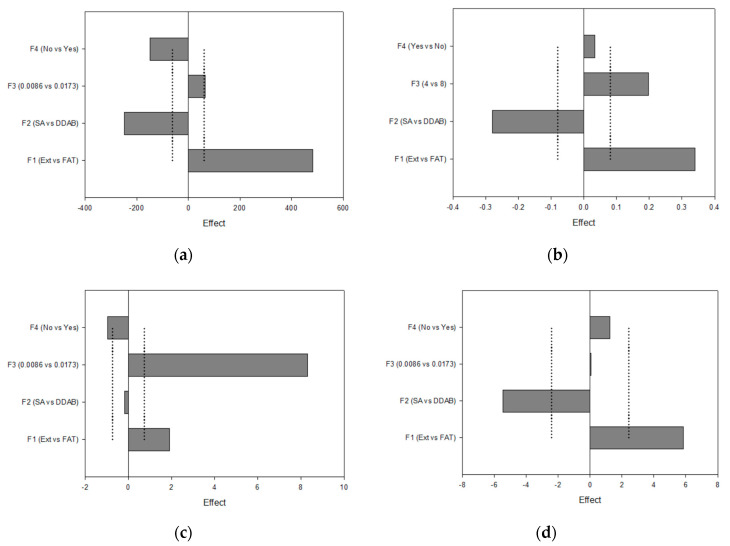
Pareto charts plotting the I-hat effects for (**a**) d_H_; (**b**) PdI; (**c**) ZP; (**d**) EE Acz.

**Figure 5 pharmaceutics-13-02010-f005:**
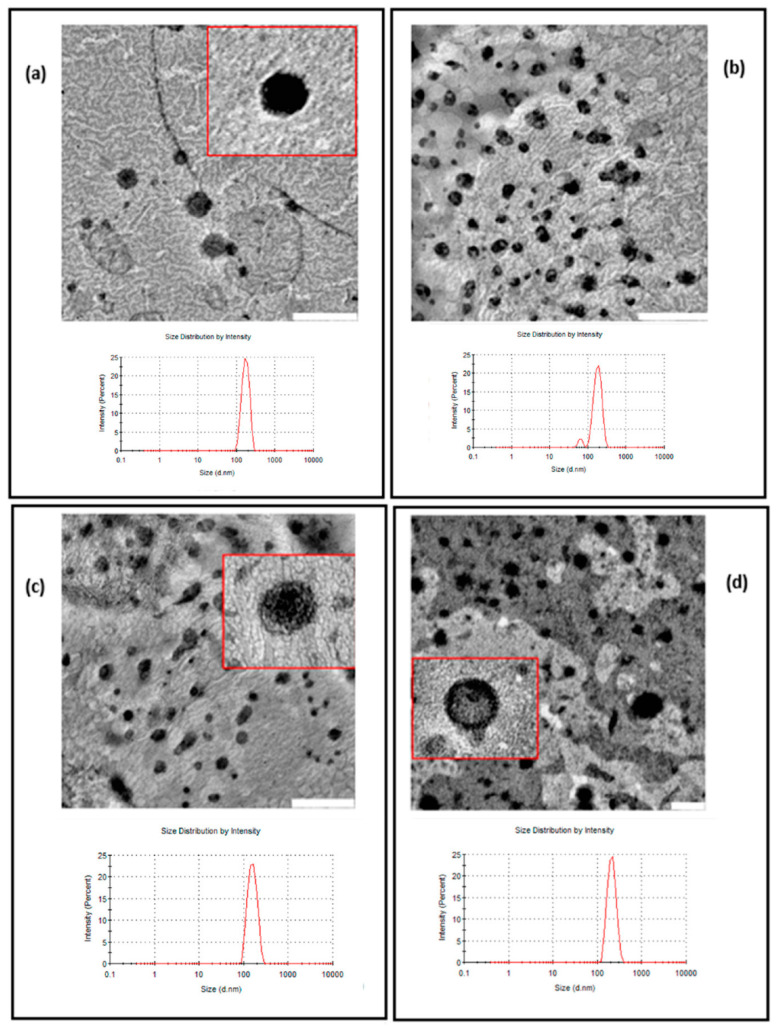
Transmission electron micrographs (TEMs) of samples and size distribution by DLS: (**a**) Empty-L; (**b**) TM-L; (**c**) Acz-L; (**d**) CLL. Scale bar for the TEM images in (**a**–**c**): 500 nm. Scale bar for the TEM image in (**d**): 200 nm.

**Figure 6 pharmaceutics-13-02010-f006:**
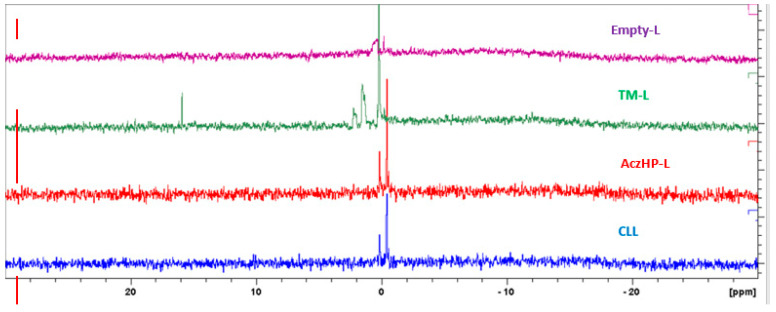
^31^PNMR spectra corresponding to empty liposomes (Empty-L), timolol-loaded liposomes (TM-L), acetazolamide–HPβCD- loaded liposomes (AczHP-L), and TM–AczHP co-loaded liposomes (CLLs).

**Figure 7 pharmaceutics-13-02010-f007:**
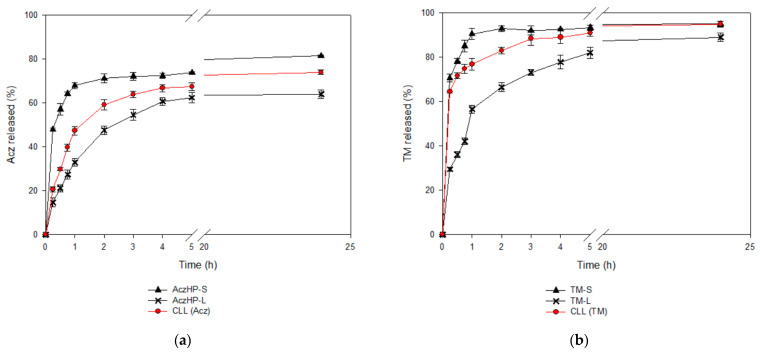
In vitro release profiles of (**a**) Acz from the co-loaded liposome formulation (CLL), Acz/HPβCD liposomes (AczHP-L), and Acz standard solution (AczHP-S); (**b**) TM from the co-loaded liposome formulation (CLL), TM liposomes (TM-L), and TM standard solution (TM-S).

**Figure 8 pharmaceutics-13-02010-f008:**
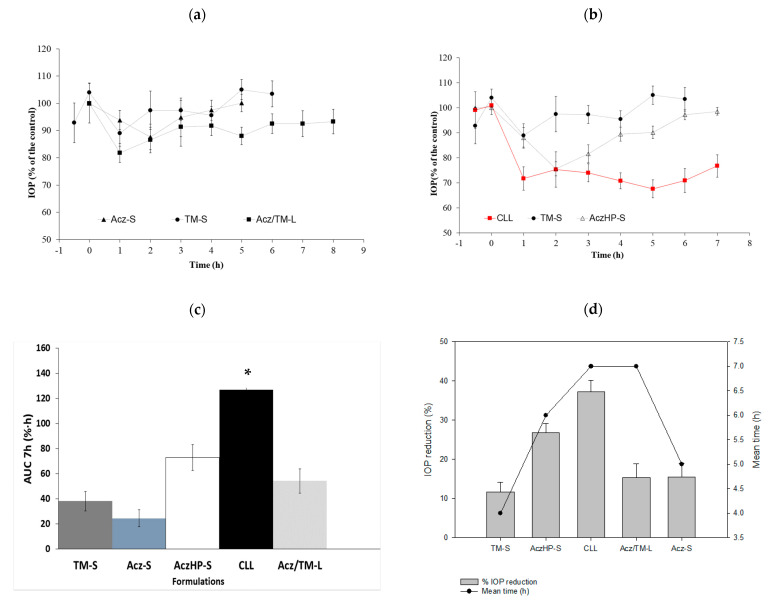
IOP profiles of (**a**): (▲) Acz-S; (●) TM-S; (■) Acz/TM-L; (**b**) (∆) AczHP-S; (●) TM-S; (■) CLL. (**c**) Area under the curve (AUC07h (%·h) ± SD) for each formulation. * Significant differences with Acz/TM-L, Acz-S, AczHP-S, and TM-S (*p*-value < 0.05). (**d**) Maximal IOP reduction (%SEM) and the mean time effect (h) of Acz and TM for each formulation.

**Table 1 pharmaceutics-13-02010-t001:** (A) Factors selected in the study with the nomenclature and their levels. FAT: freeze–thawed vesicles; SA: stearylamine; DDAB: didodecyldimethylammonium bromide. (B) L8 Taguchi orthogonal array used in this study. (−) Lower level of the factor; (+) higher level of the factor. F1: method for reducing size; F2: cationic lipid; F3: amount of cationic lipid (mmol); F4: double loading.

(**A**)				
**Factor**		**Alias**	**Level (−)**	**Level (+)**
Method for reducing size		F1	Extrusion	FAT
Cationic lipid		F2	SA	DDAB
Amount of cationic lipid (mmol)		F3	0.0086	0.0173
Double loading		F4	No	Yes
(**B**)				
**Run**	**F1**	**F2**	**F3**	**F4**
1	−	−	−	−
2	−	−	+	+
3	−	+	−	+
4	−	+	+	−
5	+	−	−	+
6	+	−	+	−
7	+	+	−	−
8	+	+	+	+

**Table 2 pharmaceutics-13-02010-t002:** Characteristics of the Acz/HPβCD complexes. S_0_: solubility of Acz in the absence of HPβCD; n: slope of the logarithmic plot; K_c_: cumulative complex stability constant; K_s_: apparent stability constant; ESR: enhanced solubility ratio; CE: complexation efficiency.

		$\log\left(S_{t}-S_{0}\right)=nlog\left[HP\beta CD\right]+log\left(K_{c}S_{0}\right)$				$K_{s}=\frac{slope}{S_{0}\left(1-slope\right)}$			
	S_0_ (mM)	N	K_c_ (M^−1)^	*r* ^2^	Slope	*r* ^2^	K_s_ (M^−1^)	ESR	CE
Acz/HP	3.49 ± 0.18	0.98 ± 0.02	50.3 ± 2.3	0.993 ± 0.004	0.1601 ± 0.0004	0.996 ± 0.001	54.8 ± 0.9	5.6 ± 0.3	0.191 ± 0.001
Acz/HP/Ch	2.57 ± 0.15	1.00 ± 0.04	41.1 ± 7.5	0.992 ± 0.006	0.1094 ± 0.0041	0.993 ± 0.006	47.9 ± 1.8	5.3 ± 0.1	0.123 ± 0.005
Acz/HP/Chems	2.94 ± 0.15	0.88 ± 0.01	93.5 ± 3.6	0.998 ± 0.006	0.1550 ± 0.0005	0.996 ± 0.001	62.5 ± 3.6	6.4 ± 0.3	0.183 ± 0.001

**Table 3 pharmaceutics-13-02010-t003:** Chemical shifts (ppm) in Acz and HPbCD in the free and complexed forms (1:1).

		δ_free_	δ_complexed_	Δδ (ppm)
	H_1_	5.0030	5.0069	0.0039
	H_2_	3.3974	3.3926	−0.0048
HPβCD	H_3_	3.9304	3.9537	0.0233
	H_4_	3.5410	3.5440	0.0030
	H_6_	3.7919	3.7890	−0.0029
Acz	–CH_3_	2.2529	2.2392	−0.0137

**Table 4 pharmaceutics-13-02010-t004:** ANOVA analysis for the responses evaluated. A *p*-value < 0.001 was considered statistically significant (*). F1: method for reducing size; F2: cationic lipid; F3: amount of cationic lipid (mmol); F4: double loading.

d_H_			
	F-ratio	*p*-Value	% Contribution
F1 (Ext vs. FAT)	63.79	<0.001	59.3 *
F2 (SA vs. DDAB)	16.85	0.003	15.7 *
F3 (0.0086 vs. 0.0173)	1.17	0.312	1.1
F4 (No vs. Yes)	5.98	0.04	5.6 *
**PdI**			
	F-ratio	*p*-Value	% Contribution
F1 (Ext vs. FAT)	9.12	0.017	24.1 *
F2 (SA vs. DDAB)	6.19	0.038	16.4 *
F3 (0.0086 vs. 0.0173)	3.08	0.117	8.1
F4 (No vs. Yes)	0.08	0.78	0.2
**ZP**			
	F-ratio	*p*-Value	% Contribution
F1 (Ext vs. FAT)	6.443	0.035	4.6 *
F2 (SA vs. DDAB)	0.062	0.81	0
F3 (0.0086 vs. 0.0173)	121.7	<0.001	87 *
F4 (No vs. Yes)	1.632	0.237	1.2
**EE Acz**			
	F-ratio	*p*-Value	% Contribution
F1 (Ext vs. FAT)	6.031	0.04	6.5 *
F2 (SA vs. DDAB)	5.281	0.051	5.7 *
F3 (0.0086 vs. 0.0173)	0	0.988	0
F4 (No vs. Yes)	0.271	0.617	0.3

**Table 5 pharmaceutics-13-02010-t005:** Selected levels for each factor, which were extracted from the Pareto analysis and for which the different responses in the sense of minimizing d_H_ (size) and maximizing PdI (polydispersity index), ZP (zeta potential), and EE Acz (encapsulation efficiency of acetazolamide) were optimized.

	d_H_	PdI	ZP	EE Acz
F1 (Ext vs. FAT)	Extrusion	Extrusion	FAT	FAT
F2 (SA vs. DDAB)	DDAB	DDAB	-	SA
F3 (0.0086 vs. 0.0173)	-	0.0086	0.0173	-
F4 (No vs. Yes)	Yes	-	No	-

**Table 6 pharmaceutics-13-02010-t006:** Characterization parameters of the optimized formulation (CLL) co-loading AczHP and TM in the same vesicle and the formulations loading TM (TM-L) and AczHP (AczHP-L) separately. The groups CLL, TM-L, and AczHP-L were compared with Empty-L (*** *p* < 0.001; * *p* < 0.05).

Batch	dH (nm)	PdI	ZP (mV)	EE (%)
CLL	179.1 ± 0.3 ***	0.107 ± 0.02	25.7 ± 2.4	80.9 ± 5.6 (TM)/70.8 ± 4.4 (Acz)
TM-L	178.9 ± 1.1 ***	0.121 ± 0.02	32.3 ± 3.8	75.3 ± 4.2
AczHP-L	216.7 ± 0.7 ***	0.124 ± 0.01	18.9 ± 2.2 *	69.5 ± 3.3
Empty-L	159.8 ± 0.4	0.132 ± 0.01	28.8 ± 3.9	-

**Table 7 pharmaceutics-13-02010-t007:** Release and permeation parameters. J: flux rate; P: permeability coefficient. *** *p* < 0.001; ** *p* < 0.01 and * *p* < 0.05 (comparing between the same formulations); + *p* < 0.05; ++ *p* < 0.01 and +++ *p* < 0.001 (comparing liposomal formulations with the standard solutions).

Formulation (Code)	t_50%_ (min)	% Permeated 24 h	Permeated Amount of Drug at 24 h (µg/cm^2^)	J (µg/cm^2^·min)	P (mm/h)
TM-S	10.6 ± 0.5	94.0 ± 5.4	149.7 ± 10.1 ***	1.1 ± 0.1 ***	0.70 ± 0.02
AczHP-S	18.3 ± 0.8	80.0 ± 6.1	1273.9 ± 22.5	9.0 ± 0.8	0.67 ± 0.09
CLL (TM)	11.7 ± 0.2	96.7 ± 4.5	153.9 ± 9.5 ***	0.8 ± 0.1 ***	0.51 ± 0.03 +++
CLL (Acz)	73.3 ± 0.4	83.6 ± 6.7	1330.9 ± 31.8	5.6 ± 0.5 ++	0.36 ± 0.04 +**
TM-L	54.0 ± 0.5	91.0 ± 5.7	144.9 ± 5.8 ***	0.9 ± 0.1 ***	0.57 ± 0.08
AczHP-L	140.2 ± 0.7	81.0 ± 4.4	1289.8 ± 41.2	5.9 ± 0.7 ++	0.38 ± 0.06 +*

## Data Availability

All data available are reported in the article.

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
