# Peer review of "Synergistic Effect of Acetazolamide-(2-hydroxy)propyl β-Cyclodextrin in Timolol Liposomes for Decreasing and Prolonging Intraocular Pressure Levels"

_pharmaceutics, 2021, doi:10.3390/pharmaceutics13122010_

Round 1

Reviewer 1 Report

This paper constitutes an excellent approach of facing the challenge of con-encapsulating more than one drug in the same particular or vesicular system to provide a synergistic effect. The paper is well presented and well documented. I consider that it can be published without any changes.

Author Response

Thank you very much for your kind response

Reviewer 2 Report

Very comprehensive work. Definitely worth publishing.

Author Response

(The authors gave the same response as above.)

Reviewer 3 Report

The manuscript by Arroyo-García et al. on combined liposomes for glaucoma may be of interest, but unfortunately, the current version does not meet the minimum requirements for a scientific article. Not only does some information remain hidden, but the manuscript contains only traces of a discussion of the results, and the Conclusions section also needs some extension. There are many typos, and some experimental results appear to have been omitted or experiments not performed at all. Although the authors' major revision may convert it scientifically acceptable, the next revision can reveal some other crucial problems.

Only the referee's few major concerns are below, as the shortcomings of the Discussion section do not allow a full scientific assessment.
- The problems start with the Title and Abstracts.
The HPßCD in the title does not contain only typos but is scientifically incorrect. HPßCD is the acronym of (2-hydroxy)propyl ßCD or (2-hydroxy)propylated ßCD. The HP pendant group has an OH functionality on the 2nd carbon, and this is why it is necessary to discriminate this version from the (3-hydroxy)propyl version. However, this latter is unauthorized as a pharmaceutical additive, and its commercial availability is also restricted.
The authors missed informing the reader about the DS (Degree of Substitution, number of HP groups/macrocycle, according to pharmacopeias) of the HPßCD. It is crucial, and see the details later.
 Line 16 contains incorrect capital letters. There is also a new version for HPßCD.

- The Keywords consist of 50 % of the title. Five authors must be more creative than copy/pasting.

- Unfortunately, although the Introduction appears comprehensive and complete, the authors felt it necessary to include additional introductory information in the first part of the Discussion section.
Lines 679-704 should be in the Introduction, as these sentences do not discuss the authors' results. After the reorganization, the Introduction section might need some shortening.

- Materials and Methods
The DS neither is mentioned in this section, which suggests that the authors have not aware the knowledge that around four different types (different DSs, ~3.5, ~4.3-4.5, ~5.2, ~6.3, and eventually DS~11, but the quality of this latter is questionable) of HPßCDs are available on the market. All of them are in the pharmacopeia accepted DS range. The authors neither seem to be aware that HPCDs can exhibit DS-dependent complexation properties for several guests. Cholesterol is a well-known example, and the authors used two cholesterol variants in their experiments! The lack of experiments with different HPßCD versions would be a minor problem, but the authors used DOE for optimizing something - which is, by the way, not mentioned in the Abstract. What is a DOE which misses a crucial experimental parameter? Did the authors know what the subject of their studies was?

- The correct version of the sentence starts in line 154 is: "... TM was performed by high performance liquid chromatography (HPLC) using a Hitachi Elite La Chrom system, equipped with a reverse-phase column, an L-2130 ..."

- The logarithmic equation and Kc definition are stimulating and might be practical, so the referee is concerned that the authors missed some mathematical consequences of the equation; namely, the parameter fitting after the logarithmic transformation of data, unavoidable leads to significantly higher error than those calculations which use untransformed values. Double logarithmic data transformations and then exponential parameter transformations cause computational errors to accumulate. Apart from this, ref. 59 in the definition of Kc contains error (above equation 1), but it is not the problem of this manuscript. To avoid misunderstandings, the "cumulative complex stability constant" is a better definition of Kc.
Moreover, the apparent stability constant is apparent precisely because it neglects complex compositions other than simple compositions (e.g., 1:1, 2:1, 1:2, and so on), assumes by definition simplified computations and various complexes. Usually, the formation of multiple CD complexes is a step-by-step process. It is also true for the HPßCD complexes because although the HPßCD concentration is much higher than that of the guest molecule, it is usually still a diluted solution for HPßCD (up to ~5 %, ~30-40 mM). The HPßCD can associate with higher concentrations (>10-15%), and the bending of the linear curve can indicate the presence of other than 1:1 (or 2:2, 3:3) complexes. The goodness of parameter fitting (correlation coefficient, r^2) can also help the solubility isotherm analysis and infer the composition of the complex.

- Section 2.5.1, the temperature in particle size and zeta-potential experiments is undefined. "Room temperature" is not an accurate definition, especially since the temperature is a factor in calculations of hydrodynamic diameter (minimal influence, because of the absolute temperature scale) and zeta-potential (higher effect). The instrument software usually displays the sample temperature (if the temperature probe is connected, of course). Another unexplained experimental setting is that although the authors correctly used the buffer for the particle size measurement, the zeta-potential experiments used water only. Significantly more ions are obviously in a buffered solution, but how can the authors conclude anything from particle size and zeta-potential experiments under different conditions? The association properties depend on the zeta-potential, which can be necessarily different in pure water and buffer.
Using the ζ alone is not the best choice for the zeta-potential nickname because the symbol conversion is sometimes unsuccessful. ZPT or something like that might be more advantageous.

- The authors recorded the 31P-NMR spectra. The addition of full-size (half-page) spectra would be advantageous to the SI because a single table with minimal significant data is weak.
Line 275 contains RMN, which is an unresolved abbreviation (of course, a typo).
The acquisition of proton-NMR spectra is also recommended for the authors, as it may also reveal the concentration dependence of the proton shifts of Acz acetyl. Not only to further confirm the complex formation but also to potentially calculate the apparent stability constant by another method. At 500 MHz, they can carry out the experiments also in H2O-D2O mixtures. According to the literature, the acetyl CH3 signal is well-separated from the CH3 peak of the HP group. From 1HNMR, even the authors themselves can calculate the (average) DS value of HPßCD and the (average) MW value - when the CoA of the HPßCD is unavailable. Set the integral value of the anomeric signals to 7, then divide the HP CH3 value by three; MW=DS*58+1135. The pharmacopeia method of DS calculation uses this method.

- Section 2.6, experimental design weakness has been mentioned before. The authors should give clues for using their HPßCD version (apart from the support of the producer).

- Figure 1. Although the authors have used the solubility isotherm data in two ways, they have failed to include the log-log isotherm plot. They used the log-log plot in the calculation of Kc.

- In line 458, the "apparent" is missing for Ks.

- Table 1.
The capital N is not in agreement with the column heading equation (lower case n).
The SD of the parameter values is missing, so are the r^2 values of the fitting. Interestingly, in other tables, they could provide the SDs. Please also tabulate the measured St and S0 values, eventually in the SI. Seeing is believing.
The values of n in rows of Acz/HP and Acz/HP/Ch are not significantly different from 1. These values confirm the almost absolute existence of 1:1 complexes. Furthermore, the 10-20% differences of Kc and Ks in these cases also demonstrate the weakness of log-log fitting and the exponential conversion to calculate K.
The authors also missed the discussion of the significantly less solubilization power of the complex in the presence of Ch and Chems.

- The contribution values in Table 3 are impressive, but their calculation method is missing. The mystic lines 481-483 are insufficient.

- Caption of Table 4 is unclear. Does this table contain the effect of various parameters for the experimental values? If yes, then please write that.

- Table 5.
From the Experimental section, the reviewer concluded that the dH and the zeta-potential are not in relation. As mentioned before, the dH in the buffer has no correlaítion with the zeta-potentials in pure water.
Furthermore, it appears that the authors calculated the dHs from lognormal distributions, but the skewness of the distribution curves of Figures 4b-4d suggests that the differences are not always significant. Or, in other words, the shown dHs need further explanation. The authors also missed the figures and values of the multimodal distributions - and, of course, their discussion, as well.
Do the authors have an idea for why the zeta potential of Acz/HP/L is significantly different from the others? Their work suggests that liposomes encapsulate the Acz/HP, i.e., the outer sphere of the liposomes does not change significantly. According to the theory, the zeta-potential shows only minimal, almost negligible, particle size dependency - which differences in these cases, looking at the dHs, are not dramatic.

- The Discussion does not discuss anything, practically, presents data in text only. As mentioned before, the first three paragraphs do not belong there.

- The Conclusion needs extension with valid conclusions. The authors used DOE, but in the Conclusion, they missed to conclude how DOE helped their work.

The reviewer proposes considering these concerns before the resubmission because a more detailed scientific evaluation of the manuscript is possible after their corrections.

Author Response

See by attachment

Reviewer 4 Report

 This paper describes the formulation design of liposomes consisting of timolol and hydroxypropyl-b-cyclodextrin-solubilized acetazolamide and its decreasing effect of intraocular pressure in vivo. The study is comprehensive, from the solubilization to biological efficacy, and contains useful information for glaucoma therapy. The following unclear point should be clarified.

  1. Section 2.2: Please add the UV wavelength used in the HPLC analysis of the drugs.
  2. Table 2: Experimental errors should be added for all figures of Table 2, because of the small difference in these parameters.
  3. Figure 1: The intrinsic solubility (So) of Acz was changed by the addition of Ch and Chems, indicating that Acz interacts with Ch and Chems, probably owing to some micelle formation or other mechanisms. Please add some comments on this matter.
  4. In vitro drug release experiments: The absolute amount of both drugs used in the release study should be added for all formulations (CLL, TM-L, AczHP-L, TM-S and AczHP-S).
  5. In vivo hypotensive efficacy: Similarly, the amount of both drugs used in the in vivo hypotension study should be described for all formulation. If the absolute amounts of the drug are different between CLL, AczHP-L and Acz-S etc., it is apparent that the biological effect changes.

Author Response

See by attachment

Round 2

Reviewer 3 Report

The manuscript of Arroyo-García is significantly improved. The authors' feedbacks correctly answered all concerns, and only two - one minor and one major issue remained.
Minor, the "hydroxy)propylβcyclodextrin" is incorrectly punctuated. The correct version is hydroxy)propyl β-cyclodextrin (if "CD" is used instead of "cyclodextrin" the dash after the beta sign is unnecessary).

Major: The referee is afraid that the authors misunderstood something about the particle size graphs. The referee mentioned the necessity of the multimodal particle size distribution charts (MSD), not the autocorrelation charts. Since the instrument software can generate distribution charts as relative percentages to the maximum or as a percentage distribution graph (sum=100%) - this is software dependent -, please insert one of these distribution graph types instead of the autocorrelation charts.

Author Response

Answers to referee’s comments have been carefully treated and they are provided below.

We expect the revised manuscript is considered suitable for publication in Pharmaceutics.

There is no conflict of interest to disclose, and no part of the manuscript has been published elsewhere.

We will be waiting to hearing from you soon.

Thank you very much in advance.

Sincerely yours,

        The Authors

Reviewer 4 Report

The paper has been correctly revised and will be acceptable for publication.

Author Response

Thank you very much for your comments

Round 3

Reviewer 3 Report

Authors' last corrections converted their manuscript into a publishable version. The referee understands their difficulties with the MSD graph, but the tabulated version is correct and scientific.